# Highly Sensitive Detection of Zika Virus Nonstructural Protein 1 in Serum Samples by a Two-Site Nanobody ELISA

**DOI:** 10.3390/biom10121652

**Published:** 2020-12-09

**Authors:** Triana Delfin-Riela, Martín Rossotti, Romina Alvez-Rosado, Carmen Leizagoyen, Gualberto González-Sapienza

**Affiliations:** 1Cátedra de Inmunología, DEPBIO, Instituto de Higiene, Facultad de Química, UDELAR, Montevideo 11600, Uruguay; triana@fq.edu.uy (T.D.-R.); martinrossotti@gmail.com (M.R.); ralvezrosado@gmail.com (R.A.-R.); 2Parque Lecoq, IMM, Montevideo 12600, Uruguay; carmenleizagoyen@gmail.com

**Keywords:** diagnosis, flavivirus, NS1, immunoassay, phage display, single-domain antibody

## Abstract

The Zika virus was introduced in Brazil in 2015 and, shortly after, spread all over the Americas. Nowadays, it remains present in more than 80 countries and represents a major threat due to some singularities among other flaviviruses. Due to its easy transmission, high percentage of silent cases, the severity of its associated complications, and the lack of prophylactic methods and effective treatments, it is essential to develop reliable and rapid diagnostic tests for early containment of the infection. Nonstructural protein 1 (NS1), a glycoprotein involved in all flavivirus infections, is secreted since the beginning of the infection into the blood stream and has proven to be a valuable biomarker for the early diagnosis of other flaviviral infections. Here, we describe the development of a highly sensitive nanobody ELISA for the detection of the NS1 protein in serum samples. Nanobodies were selected from a library generated from a llama immunized with Zika NS1 (ZVNS1) by a two-step high-throughput screening geared to identify the most sensitive and specific nanobody pairs. The assay was performed with a sub-ng/mL detection limit in the sera and showed excellent reproducibility and accuracy when validated with serum samples spiked with 0.80, 1.60, or 3.10 ng/mL of ZVNS1. Furthermore, the specificity of the developed ELISA was demonstrated using a panel of flavivirus’ NS1 proteins; this is of extreme relevance in countries endemic for more than one flavivirus. Considering that the nanobody sequences are provided, the assay can be reproduced in any laboratory at low cost, which may help to strengthen the diagnostic capacity of the disease even in low-resource countries.

## 1. Introduction

The Zika virus (ZV) is an arthropod-borne virus, isolated for the first time in Uganda in 1947 [1], that belongs to the Flaviviridae family. Nevertheless, ZV remained almost unnoticed for fifty years, until, in 2015, it was introduced in Brazil and, shortly after, spread all over the continent [2,3]. Thus, in 2016, the World Health Organization (WHO) declared the ZV epidemic as an international public emergency. This virus represents a major threat due to some singularities not common among other flaviviruses. In particular, apart from the bite of infected *Aedes* spp. mosquitoes, ZV can be vertically transmitted during pregnancy or breastfeeding, as well as spread through sexual contact or blood transfusion [4]. Moreover, although symptoms are usually mild, complications are severe. In particular, a ZV infection carries the risk of Guillain-Barré Syndrome and several adverse pregnancy and congenital outcomes, collectively known as congenital Zika syndrome [5,6]. Although the intensity of the epidemic decreased in the past years, it is important to have a rapid and reliable diagnostic test to monitor the transmission and help to contain eventual outbreaks. Until now, a ZV diagnosis mainly depends on the detection of viral RNA in patients’ sera by quantitative polymerase chain reaction (qPCR). RNA quantification is highly specific, but its efficiency drops after seven days of the onset of symptoms [7]. At the same time, the process is laborious and requires expensive equipment and specialized operators. On the other hand, when the viremia is about to disappear, the antibody response begins to become evident; therefore, during the convalescent phase, the detection of immunoglobulin M (IgM) and later immunoglobulin G (IgG) by immunoassays are preferred. However, due to extensive shared similarities between ZV and other flaviviruses, such as Dengue (DV), West Nile (WN), and Yellow fever (YF), considerable immunological cross-reactivity has been observed [8]. There is plenty of evidence of false-positive results for patients living in endemic areas for more than one Flavivirus [9,10]. Consequently, this constitutes a major drawback in the use of serology tests.

Nonstructural protein 1 (NS1) is a glycoprotein involved in Flavivirus infection, and its secreted form is demonstrated to be highly immunogenic; therefore, it might be used as a diagnostic biomarker of the disease [11]. It is known, from other flavivirus infections, that NS1 appears in blood concomitantly with viremia and circulates in large amounts, even up to twelve days after fever onset; hence, it is an indicator of ongoing or recent infection [11,12,13,14]. Even more, the detection of this protein can be carried out through a simple capture ELISA [15]. Nonetheless, the evaluation of several presently available DV serological tests based on the NS1 detection cross-react with the ZV protein, probably due to the important structure similarity between them [16]. In addition, little progress has been made on the Zika NS1 (ZVNS1) detection, and the evaluation of cross-reactivity is limited.

In the past years, nanobodies (Nbs), the recombinant fragment derived from the variable domain (VHH) of camelid heavy-chain-only antibodies, have been recognized as versatile and advantageous diagnostic reagents [17,18]. Their monodomain nature facilitates the construction and expression of highly diverse libraries using phage displays. Once isolated, the Nbs are easily expressed in soluble form in the periplasm of *Escherichia coli*, with much higher expression levels than those of conventional antibodies fragments [19]. Furthermore, due to their high stability and robustness, outstanding stability, low-cost production in bacteria, and indefinite reproducibility from its known sequence, they provide an improvement in the robustness and lower test costs [18,20]. Previously, our group described a high-throughput methodology for the selection of pairs of nanobodies that allowed the development of sensitive sandwich immunoassays [21]. This permitted us to generate a highly sensitive, low-cost sandwich Nb-ELISA for the specific detection of ZVNS1 as an early marker for the diagnosis of Zika acute infection.

## 2. Materials and Methods

### 2.1. Materials

Flaviviruses’ nonstructural protein 1 and mouse anti-ZVNS1 monoclonal antibody were purchased from The Native Antigen Company (Oxford, OX, UK). Anticoagulant citrate dextrose solution (ACD), histopaque-1077, Tween 20, polyethylene glycol 8000 (PEG), bovine serum albumin (BSA), IPTG (isopropyl β-d-1-thiogalactopyranoside), d-biotin, trypsin from bovine pancreas, 3,3′,5,5′-tetramethylbenzidine (TMB), and other common chemicals were purchased from Sigma-Aldrich (Mississauga, MO, USA). TRIZOL reagent was from Invitrogen (Carlsbad, CA, USA). M-MuLV Reverse Transcriptase, random primer mix, Taq polymerase, helper phage M13KO7, and all restriction enzymes were purchased from New England Biolabs (Ipswich, MA, USA). The pComb3X vector was a kind gift from Dr. Carlos Barbas, The Scripps Research Institute (La Jolla, CA, USA). *E. coli* ER2738 electrocompetent cells and recovery medium were purchased from Lucigen Corporation (Middleton, WI, USA). The primers used for library construction were from Integrated DNA Technologies (Coralville, IA, USA). Plasmid extraction, PCR clean up, and gel extraction kits were acquired from Qiagen (Germantown, MD, USA). Bacterial protein extraction reagent (BPER), NHS (*N*-Hydroxysuccinimide)-biotin, and streptavidin peroxidase were purchased from Thermo Fisher (Rockford, IL, USA). ELISA strips and plates were from Greiner Bio-One (Monroe, NC, USA). Anti-hemagglutinin mAb (3F10) peroxidase conjugate was from Roche (Madison, WI, USA). His-Pur Ni-NTA chromatography columns were from GE Health Care (Pittsburgh, USA).

### 2.2. Llama Immunization and Library Construction

The use of a three-year-old female llama (*Lama glama*) for this study was approved by the authorities of the Zoológico Parque Lecocq, Montevideo, Uruguay. All the procedures were carried out by the veterinarians of the zoo following the protocol approved by the Comisión de Ética en el Uso de Animales del Parque Lecocq (CEUA), protocol number CEUA-1-141107. The animal was immunized by subcutaneous injection with 150 µg of ZVNS1 in incomplete Freund adjuvant, as described previously [22,23,24]. Three additional boosters were performed every 3 weeks in the same conditions. Fifteen days after the final booster, 150 mL of blood were collected in bags containing sodium citrate as the anticoagulant. Peripheral mononuclear cells were obtained by centrifugation on histopaque-1077 gradients according to the manufacturer’s recommendations. Total RNA from 10^7^ cells was extracted using TRIZOL, and 10 µg of it was reverse-transcribed using the M-MuLV Reverse Transcriptase enzyme and a random primer mix. Then, the genes encoding the variable domain of the heavy chain of conventional antibodies (VH) and heavy-chain-only antibodies (VHH) were PCR-amplified using VH1, VH3, and VH4 as forwards primers and JH as the reverse primer, as previously described [23]. Sfi sites were introduced during amplification, permitting to clone the fragments into the phagemid plasmid pComb3X. VHH/VH-digested fragments were separated in 1% agarose, purified by gel extraction, and 1.2 µg were ligated overnight at 16 °C with 1.0 µg of SfiI-digested pComb3X. Then, the ligation mix was concentrated and desalted by ethanol precipitation, resuspended in 25 µL of water, and electroporated in *E. coli* ER2738. Cells were allowed to recover 1 h in recovery medium and were then inoculated to 10 mL of LB (Luria-Bertani) broth containing 100 µg of ampicillin and incubated for 2 h with agitation at 37 °C. *E. coli* cells were superinfected with M13KO7 helper phage for 30 min without agitation, and then, kanamycin was added at a concentration of 50 µg/mL and cultured overnight (ON) with shaking in the same conditions. Next day, the supernatant was harvested by centrifugation, and phages were precipitated twice with 0.2 volume of 20% polyethylene glycol 8000 in 2.5-M NaCl and resuspended in phosphate-buffered saline (PBS) containing 3% BSA, 0.3% Tween 20, 10%, glycerol, and 150-mM L-arginine. The VH/VHH phage library was titrated by infection of *E. coli* and stored at −80 °C.

### 2.3. Purification of Immune Llama Immunoglobulins

The total fraction of immunoglobulins were obtained from the serum of the immune llama using a protein A column from GE healthcare (Piscataway, NJ, USA), as described in [17]. After dialysis against PBS, the immune llama IgG (ill-IgG) were stored at −20 °C.

### 2.4. Panning Strategies for the Selection of ZVNS1-Specific Antibodies

For panning, ZVNS1 was immobilized on high-binding ELISA strips using three different strategies: (A) wells coated with 100 μL of ZVNS1 (1 μg/mL) in PBS, (B) wells coated with 100 µL of streptavidin (1 μg/mL) in PBS, followed by incubation with 100 μL of biotinylated ZVNS1 (1 μg/mL) in PBS, and (C) wells coated with 100 μL of ill-IgG (10 µg/mL) in PBS, followed by incubation with 100 μL of ZVNS1 (1 μg/mL) in PBS-0.1% Tween 20 (PBST). After each coating step (ON, 4 °C), the strips were blocked with PBS-1% BSA for 1 h at room temperature (RT) and then washed with PBST. For panning, wells were loaded with 100 μL of the 1/100 diluted antibody library (1 × 1011 colony-forming units, cfu) and incubated for 2 h with agitation at 4 °C. Unspecific phages were eliminated by 10 times washing with PBST, and the bound phages were eluted by incubation for 30 min at 37 °C with 100 μL/well of 10-mg/mL trypsin. Finally, the phages were collected and used for titration and subsequent amplification in ER2738 *E. coli* for an additional round of selection. Three rounds were performed in total.

### 2.5. Nanobody Expression

A culture 96-deep-well plate from Greiner Bio-One (Monroe, NC, USA) was prepared to produce supernatants from 24 clones randomly picked from the outcome of each of the three immobilization strategies used for panning (A, B, and C). To this end, 72 individual colonies were inoculated into 0.5 mL of Super Broth (SB)-ampicillin in a 96-deep well block, grown at 37 °C until an optical density (OD) of 1.0 AU, and then, the expression of the secreted Nb was induced by the addition of IPTG at 1-mM final concentration. The 96-well culture plate was incubated ON at 37 °C. The next day, the block was centrifuged at 1200× *g* for 20 min, and the supernatants were collected in a fresh 96-deep-well plate (“master plate”), which was a source of Nb clones throughout the study.

### 2.6. ELISA Method for Selection of Capturing Nanobodies

The ELISA plates were coated as described in panning coating. After blocking PBS-1% BSA and washing, the plates were incubated with 100 µL of the Nb supernatant. The binding of Nbs was detected using anti-HA monoclonal antibody conjugate to peroxidase (3 ng/mL). After washing, the peroxidase activity was developed by the addition of 100 μL/well of substrate solution (0.4 mL of 6 mg of TMB in 1 mL of DMSO + 0.1 mL of 1% H_2_O_2_ in water, in a total of 25 mL of 0.1-M acetate buffer, pH 5.5) and incubated at RT for 15 min. The enzyme reaction was stopped by the addition of 50 μL of 2-N H_2_SO_4_, and the absorbance was read at 450 nm on a Fluostar Optima Reader (BMG, Ortenberg, Germany).

### 2.7. Large-Scale Production of Biotinylated and HA-Tagged Nbs

The Nb genes were cloned into a pINQ-BtH6 vector and transformed into BL21(DE3) overexpressing the BirA biotin ligase of *E. coli*, as described previously [21,25]. Individual colonies were then used to inoculate 200 mL of LB containing 50-μg/mL kanamycin, 35-μg/mL chloramphenicol, and 0.04% of L-arabinose in shaking flasks. When the optical density at 600nm (OD_600_ nm) ≈ 0.6 AU, the expression of Nbs was induced with IPTG at 10 µM and grown ON at 37 °C. The next day, the bacteria was harvested, and the pellet was resuspended in 10 mL of PBS supplemented with 100 μM of D-biotin, lysed by sonication at 50% amplitude during 15 min on ice, and finally, subjected to post-biotinylation by incubation for 2 h at 37 °C [24]. After centrifugation (18,000× *g*), the soluble fraction was purified on Ni-NTA columns according to the manufacturer’s instructions. The eluted fractions were PBS-dialyzed, and the biotinylated Nbs (BtNbs) were spectrophotometrically quantified (Abs 280 nm) and kept at −20 °C until use.

HA-tagged nanobodies were produced in a similar fashion after cloning in the pINQ-H6HA vector and transformed into *E. coli* BL21 (DE3) cells. Individual colonies were used to inoculate 500 mL of LB containing 50-μg/mL kanamycin in shaking flasks. When OD_600_ nm ≈ 0.6 AU, the expression of the HA-Nbs was induced as described above. The cell pellet was then resuspended in 10 mL of PBS, lysed by sonication, and purified on Ni-NTA columns. The eluted fraction was PBS-dialyzed, quantified at 280 nm, and stored at −20 °C.

### 2.8. Pairwise Selection of Nanobodies

ELISA plates were coated ON with 100 μL/well of 1-μg/mL streptavidin, blocked with PBS-1% BSA for 1 h at RT, and then dispensed with 100 μL/well of 1-μg/mL purified capturing BtNb. After washing, the plates were incubated with 100 μL of different concentrations of ZVNS1 in PBS (0, 2.0, and 20 ng/mL) for 1 h at RT. After washing, each plate was incubated with a 1/10 dilution in PBS of the “master plate” Nbs supernatants. Binding of the secondary antibody was detected by the addition of 100 µL of 3-ng/mL anti-HA monoclonal antibody conjugate to peroxidase.

### 2.9. Nanobody Sandwich ELISA for the Detection of ZVNS1 in Serum Samples

Deidentified normal serum samples available in our laboratory from previous research work were spiked with two different known concentrations of ZVNS1 (1.5 and 4.5 ng/mL) and subjected to quantification with the nanobody sandwich ELISA. To this end, streptavidin-coated plates (0.2 ng/well) were blocked as described above and then dispensed with 100 μL of capturing BtNb (2 μg/mL). After washing, 100 μL of ZVNS1 standards or spiked samples were loaded and incubated for 1 h at RT. After washing, the purified detecting Nb was added (100 μL, 1 μg/mL) and incubated for 1 h at RT. The binding of the Nb was detected by the addition of 100 μL of 3-ng/mL anti-HA peroxidase conjugate. After washing, the peroxidase activity was developed as described above.

## 3. Results and Discussion

### 3.1. To Promote a Broad Representation of the NS1 Epitopes, Different Antigen Immobilization Strategies Were Used to Pan the Nb Library

A llama was immunized four times with 150 µg of ZVNS1, and the antibody response was followed by serum titration. The antibody titer rose rapidly after the primer and first booster and did not change significantly afterwards (Appendix A). After the fourth immunization, a VHH/VH library of 3 × 10^8^ transformants was generated from 10^7^ blood mononuclear cells. In order to maximize the recovery of Nbs defining different epitopes on the ZVNS1 antigen, the library was panned on microtiter wells with the antigen immobilized in different ways. Condition A: ZVNS1 passively absorbed into ELISA wells, condition B: biotinylated ZVNS1 immobilized on streptavidin-coated wells, and condition C: ZVNS1 captured on ill-IgG-coated wells. After three rounds of selection, a 96 deep-well culture master plate was prepared using 72 individual clones obtained from each of the three panning conditions. We next tested the reactivity of 24 Nb clones from each panning strategy with ZVNS1 immobilized using the conditions A, B, and C (Figure 1). Most of them reacted in a similar fashion with ZVNS1, regardless of the condition used for its immobilization, but a few showed differential reactivity (e.g., 4, 18, 30, etc.), providing a first evidence of the diversity of the selected clones. Based on the intensity of their readouts and the effect of the ZVNS1 immobilization method on their reactivity, an initial panel of 34 clones were submitted for sequencing, resulting in 22 (65%) unique sequences (Appendix A). The fact that 65% of the sequences were unique and most of them unrelated suggests that a large number of epitopes were represented in this initial nanobody panel. Curiously, among these 22 clones, seven (32%) corresponded to VH domains with the characteristic GLEW motif in framework 2 and the frequent Trp to Arg substitution in framework 4 found in the VH of heavy-chain-only antibodies that bear this variable domain. The significance of this finding is unknown, but this is indeed a larger-than-usual frequency of soluble VH, since they only account for up to 10% of heavy-chain-only antibodies in llamas [26].

Considering that serum is the main matrix for the detection of circulating ZVNS1, we next tested the ability of these 22 unique Nbs to recognize ZVNS1 in the presence of a human pool of Zika-negative sera (50% in PBS) (Appendix A). Although all the clones performed better in the absence of the serum, a considerable number of them maintained high reactivity in the presence of this matrix. We also tested their capacity to react with limited amounts of ZVNS1 (Figure 2). In this case, differences between tested Nbs were notorious. This provides a simple but useful criterion to limit their selection, because highly reactive clones are those that possess high levels of expression and/or high affinity, which are both advantageous features.

After this initial screening using crude supernatants, we narrowed the selection to nine of the most promising Nbs and cloned them into the pINQ-BtH6 vector for further characterization. This vector allows the high-yield expression of soluble nanobodies, as well as their enzymatically site-specific biotinylation in a 15-mer biotin-acceptor-peptide (BtAP) tag [24]. Every selected clone was produced in large scale and purified, and identical concentrations were then titrated against a fixed amount of ZVNS1 (100 ng/well) to rank their relative affinities. The Nb concentration values causing 50% of signal saturation (SC_50_) were then used as an estimator of their relative affinity for the antigen (Figure 3). All nanobodies reacted with ZVNS1 at a low concentration, with similar SC_50_ in the 1.5–8.2 ng/mL range.

### 3.2. Most Capture Nbs Had Negligible Cross-Reactivity with Other Flavivirus NS1 Proteins

Due to high-sequence conservation among flavivirus NS1 proteins (47–57% identity [27]), the undesired cross-reactivity of the Nbs might give a place to the false-positive diagnostic results, which is evident from the high degree of cross-reactivity of the sera from flavivirus-infected patients when NS1 is used as the antigen for serology. Therefore, we assayed the reactivity of the Nbs against the NS1 protein of Yellow fever, Dengue type 1, West Nile, and Saint Louis viruses (Figure 4). With the exception of Nb38 that showed a small degree of cross-reactivity with the NS1 from Yellow Fever, Dengue, and West Nile viruses, the OD signals obtained for the rest of the Nbs were negligible and similar to that obtained against BSA. The fact that most Nbs define highly Zika virus-specific epitopes is of high diagnostic relevance, since many countries are endemic for more than one flavivirus infection.

### 3.3. Selection of Best Nanobody Pair for the Detection of ZVNS1

A high-throughput approach was applied for the identification of the detection antibodies. First, based on their relative affinity, its performance in the serum, and yields of expression, we selected four capture-Nb candidates (BtNB22, BtNb246, BtNb278, and BtNb32) to screen for the best detection antibodies. These four BtNb were individually immobilized on four streptavidin-coated microtiter plates, which, in turn, were used to capture a limited amount of ZVNS1 (2 ng/well). Then, the captured ZVNS1 was detected using the 72 Nb supernatants from the master plate, prepared as described above (Figure 5). Except for a few exceptions, the working pairs were essentially the same for the four capture Nbs, suggesting that they may target overlapping epitopes on ZVNS1. Interestingly, the Nbs panned on the ZVNS1 directly adsorbed on the wells (condition A) were less efficient at forming pairs, suggesting that some of them react to denatured epitopes. In general, Nb32Bt formed a larger number of productive pairs and outperformed the others in terms of higher readouts. For that reason, Nb32 was chosen as the capture antibody, and sixteen of the detection Nbs producing the highest signals were selected to optimize the assay.

Sixteen of the detecting Nb clones that produced the highest readouts in combination with BtNb32 were sequenced, and 10 unique sequences were obtained (Appendix A), three of which (E1, A7, and B8) also appeared in the capture Nb panel shown in Appendix A (38, 224, and 210). The Nb genes of these 10 clones were transferred to the pINQ-H6HA vector for high-yield expression of the soluble nanobody fused to a HA tag (HANb). The purified HA-tagged Nbs were tested for cross-reactivity with other flavivirus NS1, as described above (Figure 6). Except for clone A7, neither of the Nbs displayed significant reactivity with any unspecific NS1 proteins. Considering that, the catching Nb is also devoid of cross-reactivity with these proteins; this result provides an additional layer of safety to warrant the specificity of the sandwich assay.

To further select the Nb partner to be used with Bt32 as the capture Nb, individual calibration curves were done for each of the selected secondary antibodies. SC_50_ values were used as indicators of assay sensitivity (Figure 7). Based on these results, the HANbD6 was chosen as the detection Nb.

### 3.4. Development of a Nano-Sandwich ELISA for Quantification of ZVNS1

Based on the overall results of the pairwise screening, the pair BtNb32/HANbD6 (32/D6) was chosen to develop the ZVNS1 32/D6 assay (Figure 8). The analytical parameters of the sandwich ELISA in saline buffer were determined. The extended range of the calibration curve was 0.20–200 ng/mL, with a linear range of 0.20–6.25 ng/mL, showing the high sensitivity achieved. Despite the fact that Nb32 and NbD6 did not show cross-reactivity with other NS1 proteins, we tested the specificity of the assay in solution against the panel flavivirus NS1 proteins (Appendix A). As expected, no cross-reactivity was observed.

To evaluate the possible interference of the matrix, identical calibration curves were also run using 1/10 and 1/2 dilutions of a pool of healthy serum samples. As expected, considering that human sera was used during the selection of the nanobodies, no significant interference of this matrix was noticed, even using a high serum concentration. To further analyze this point, individual serum samples from healthy donors were enriched with known concentrations of ZVNS1. There is scarce information of the occurrence of ZVNS1 in the blood of Zika-infected patients. Nevertheless, while the mean levels of circulating Dengue1 NS1 antigen were found to be about 120 ng/mL, the mean value reported for the Zika virus was at least four times lower (30 ng/mL), consistent with a lower viremia level [28,29]. Based on these values, and in order to validate the use of the assay to detect trace amounts of the antigen, the ZVNS1-32/D6 test accuracy was established by analyzing the recovery of the antigen from 27 healthy donors’ sera spiked with 1.5 or 4.5 ng/mL of ZVNS1 (Table 1). The recoveries for all the samples were within the recommended range by the SANTE guidance document for method validation of the European Union [30]. This showed that the ZVNS1-32/D6 ELISA accurately detects ZVNS1 in different serum samples in the low-ng/mL range. The precision of the method was also assessed by spiking the serum with 0.80, 1.6, and 3.1 ng/mL of ZVNS1. The sample was analyzed five times in the same day (intraday precision) (Table 2) or five times on five different days (interday precision) (Table 3). In both cases, the percent coefficient of variation (CV%) was lower than the acceptance criteria (CV < 20%) recommended by the SANTE guidelines.

## 4. Conclusions

Owing to its sensitivity and high specificity, the molecular diagnosis of Zika virus infections is key to the management of the disease. Alternatively, the detection of circulating ZVNS1 represents a simpler affordable option for acute diagnosis of the disease, but due to the extended similarity of flavivirus antigens, it requires a highly specific and sensitive antibody pair [16]. In this work, we generated a large library from a hyperimmunized llama and used a two-step high-throughput selection process geared to optimize the final sensitivity of the assay. The limit of quantification of the test in the serum, 0.80 ng/mL, was established experimentally by studying its recovery (accuracy) and reproducibility with a panel of spiked serum samples. This sensitivity compared favorably with other well-established antibody-based assays for NS1, such as the one described by Bosch et al. that was able to detect up to 18 ng/mL [28]. Moreover, Nb32 and NbD6 are fully devoid of cross-reactivity with other flaviviruses, which is of paramount importance for their potential diagnostic applications in countries that are endemic for coexisting flaviviruses. Considering that these Nbs can be fully reproduced at very low costs from their amino acid sequences (provided in this study), the ZVNS1-32/D6 ELISA could be replicated in any laboratory and be the basis for an affordable tool for the early diagnosis of ZIKV. Nevertheless, a validation process with relevant patient serum samples should be conducted locally in order to demonstrate its diagnostic value.

## Figures and Tables

**Figure 1 biomolecules-10-01652-f001:**
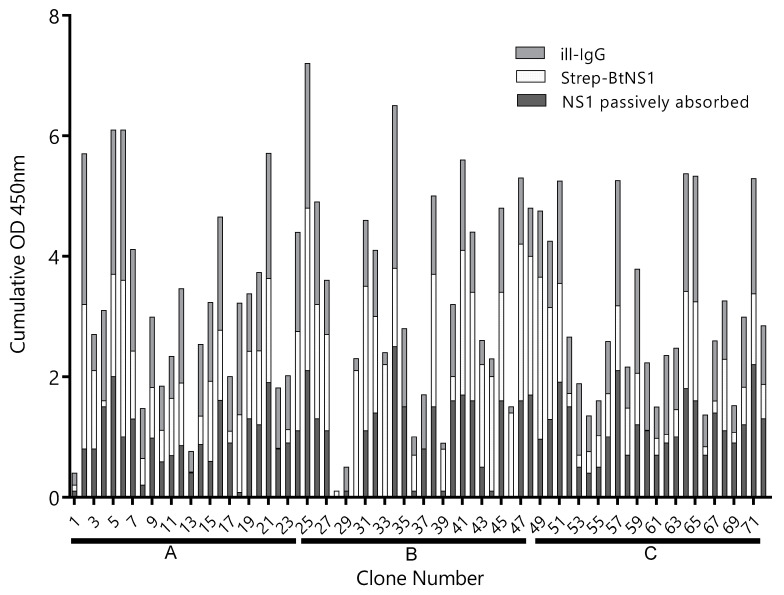
Reactivity of clones selected by different strategies. The supernatant of 24 Nb clones selected using the A, B, and C panning conditions—clones 1–24, 25–48, and 49–72, respectively—were tested by ELISA on plates coated with Zika virus NS1 (ZVNS1) immobilized using the A (dark gray), B (light gray), or C (gray) strategies. ill-IgG: immune llama immunoglobulin G and BtNS1: biotinylated NS1. Optical density at 450nm (OD450nm) was measured.

**Figure 2 biomolecules-10-01652-f002:**
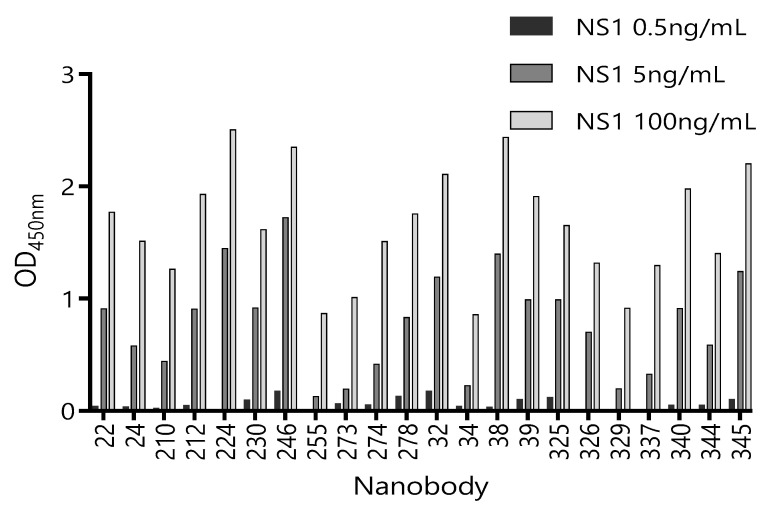
Reactivity of 22 Nb clones against different concentrations of ZVNS1. The supernatant of each clone was exposed to three different concentrations of ZVNS1 (0.5 ng/mL, 5.0 ng/mL, and 100 ng/mL) passively absorbed in the ELISA wells. Measurements were done by duplicates.

**Figure 3 biomolecules-10-01652-f003:**
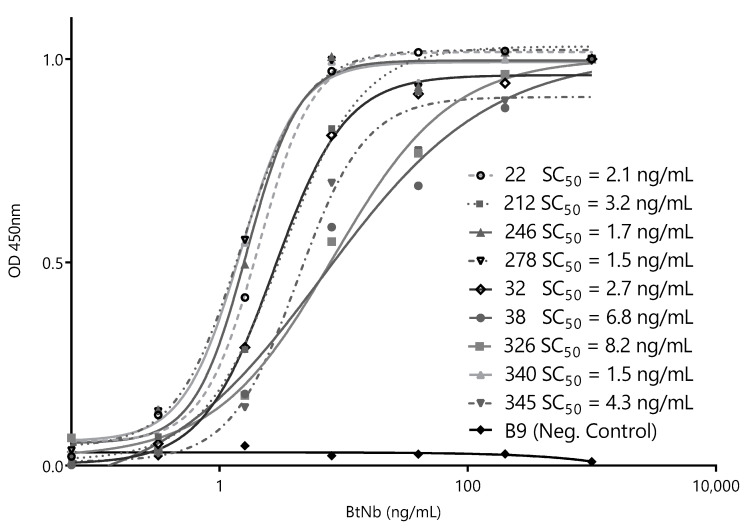
Titration curves of the selected capture nanobodies. Decreasing concentrations of each nanobody were exposed to a fixed concentration of ZVNS1 (100 ng/well) directly adsorbed on the ELISA well. Measurements were done by triplicates. SC_50_: 50% signal saturation and BtNb: biotinylated nanobodies.

**Figure 4 biomolecules-10-01652-f004:**
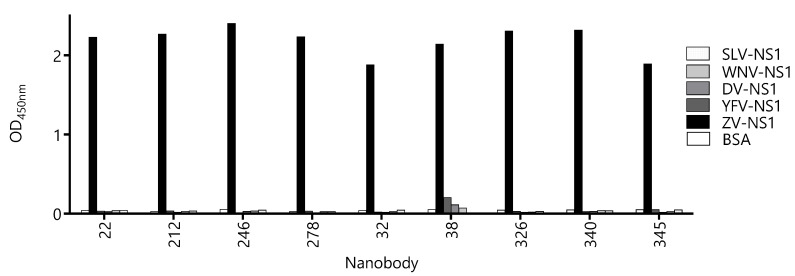
Evaluation of the cross-reactivity of the capture nanobody candidates against different flavivirus NS1 proteins. A fixed concentration of antibody was loaded into wells coated with 100 ng of NS1 from Zika virus (ZV), Yellow Fever virus (YFV), Dengue type 1 virus (DV), West Nile virus (WNV), Saint Louis Encephalitis virus (SLV), and bovine serum albumin (BSA) as a negative control. Measurements were done by triplicates.

**Figure 5 biomolecules-10-01652-f005:**
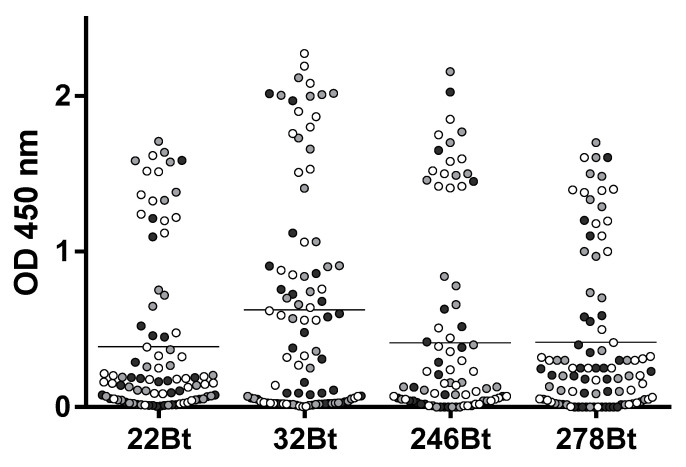
Pairwise screening of detection Nbs. The four capture Nb candidates (BtNB22, BtNb246, BtNb278, and BtNb32) were used to capture ZVNS1 (2 ng/well) and were assayed against the 72 master plate Nb supernatants. Black, gray, and white are used to denote detection Nbs selected with the A, B, and C panning strategies, respectively. The horizontal lines represent the mean OD values.

**Figure 6 biomolecules-10-01652-f006:**
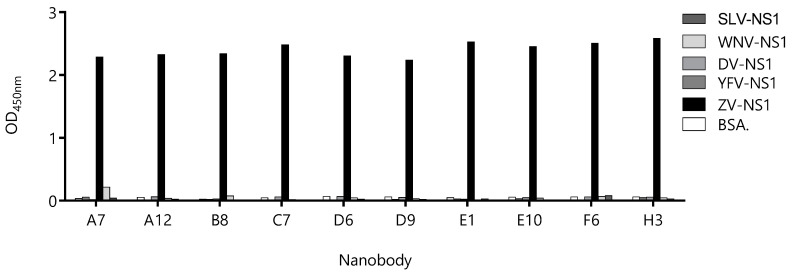
Evaluation of the cross-reactivity of the detection-nanobody candidates against different flavivirus NS1 proteins. Nanobodies were exposed to wells sensitized with 1.0 µg of NS1 from Zika Virus (ZV), Dengue Virus (DV), Yellow Fever Virus (YFV), West Nile Virus (WNV). Bovine serum albumin (BSA) was used as the negative control.

**Figure 7 biomolecules-10-01652-f007:**
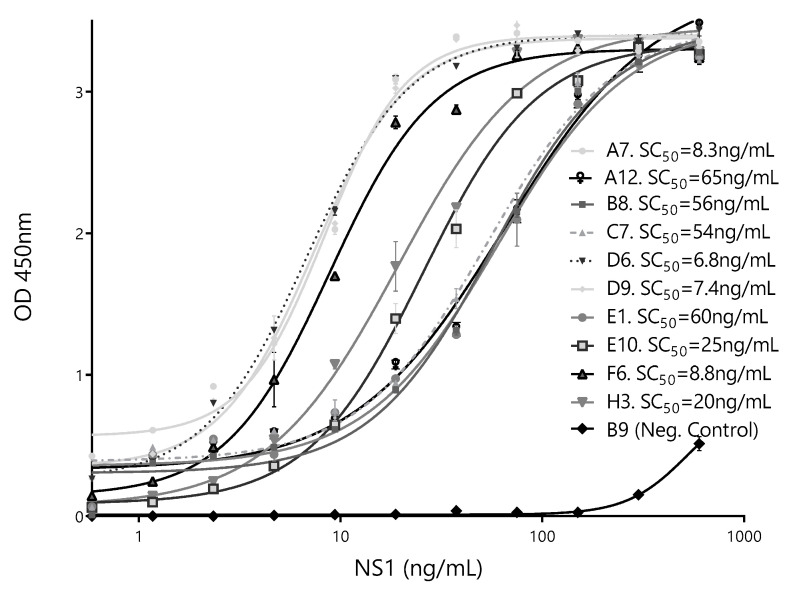
Calibration curves of the best-performing pairs. The sandwich ELISA was performed using BtNb32 as the capture antibody in combination with ten detection clones. The NbB9 with specificity for human hemoglobin was used as a negative control. The concentration corresponding to 50% of the maximum readout, SC_50_ (ng/mL), for each different pair is shown. Measurements were done by triplicates.

**Figure 8 biomolecules-10-01652-f008:**
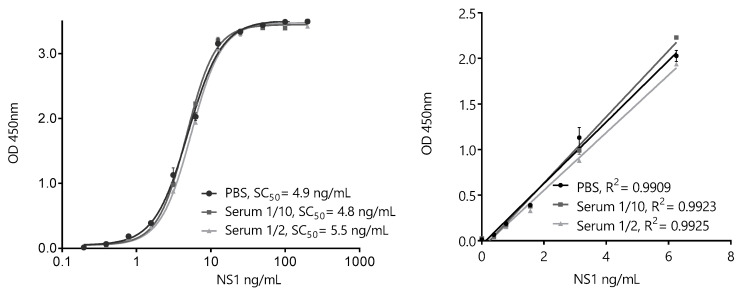
ZVNS1-32/D6 nanobody sandwich ELISA for the detection of ZVNS1. Extended (left) and linear (right)-range calibration curves performed in phosphate-buffered saline (PBS) or serum dilutions. The results are the average values of triplicate measurements, and the error bars represent the standard deviation.

**Table 1 biomolecules-10-01652-t001:** Analysis of recovery in ZVNS1 spiked serum samples.

	Unspiked	Spiked ZVNS1 1.50 ng/mL		Spiked ZVNS1 4.50 ng/mL	
Sample	ZVNS1 ng/mL	ZVNS1 ng/mL	% Recovery	ZVNS1 ng/mL	% Recovery
1	<LOQ	1.18 ± 0.10	78	4.62 ± 0.13	103
2	<LOQ	1.35 ± 0.03	90	3.54 ± 0.04	79
3	<LOQ	1.55 ± 0.01	103	4.35 ± 0.09	97
4	<LOQ	1.43 ± 0.19	95	3.97 ± 0.12	88
5	<LOQ	1.36 ± 0.01	90	4.20 ± 0.01	93
6	<LOQ	1.32 ± 0.20	88	3.68 ± 0.23	82
7	<LOQ	1.33 ± 0.16	89	4.14 ± 0.05	92
8	<LOQ	1.66 ± 0.09	110	5.26 ± 0.07	117
9	<LOQ	1.57 ± 0.12	105	4.96 ± 0.13	110
10	<LOQ	1.08 ± 0.05	72	4.02 ± 0.15	89
11	<LOQ	1.27 ± 0.10	84	3.91 ± 0.01	87
12	<LOQ	1.77 ± 0.09	118	4.10 ± 0.10	91
13	<LOQ	1.17 ± 0.21	78	3.96 ± 0.02	88
14	<LOQ	1.53 ± 0.14	102	3.46 ± 0.15	77
15	<LOQ	1.29 ± 0.18	86	4.58 ± 0.10	102
16	<LOQ	1.30 ± 0.06	87	4.20 ± 0.03	93
17	<LOQ	1.30 ± 0.30	86	4.17± 0.21	93
18	<LOQ	1.54 ± 0.11	102	4.77 ± 0.04	106
19	<LOQ	1.31 ± 0.02	87	4.49 ± 0.13	100
20	<LOQ	1.82 ± 0.09	121	4.22 ± 0.20	94
21	<LOQ	1.60 ± 0.14	107	4.19 ± 0.05	93
22	<LOQ	1.38 ± 0.25	92	4.47 ± 0.19	99
23	<LOQ	1.48 ± 0.06	99	4.62 ± 0.01	103
24	<LOQ	1.14 ± 0.10	76	4.19 ± 0.10	93
25	<LOQ	1.34 ± 0.19	89	3.88 ± 0.31	86
26	<LOQ	1.30 ± 0.01	86	3.14 ± 0.04	70
27	<LOQ	1.54 ± 0.27	103	3.89 ± 0.11	86

Samples were measured in triplicates. The mean value ± the standard deviation is shown. Spiked-ZVNS1: Zika virus NS1 and LOQ: limit of quantification.

**Table 2 biomolecules-10-01652-t002:** Intraday precision of the test with ZVNS1-spiked serum.

Assay	Unspiked Serum	Spiking 0.80 ng/mL		Spiking 1.60 ng/mL		Spiking 3.10 ng/mL	
		Measured (ng/mL)	Recovery %	Measured (ng/mL)	Recovery %	Measured (ng/mL)	Recovery %
1	<LOQ	0.81	102	1.51	94	3.16	102
2	<LOQ	0.81	102	1.52	95	3.18	102
3	<LOQ	0.69	86	1.47	92	3.21	104
4	<LOQ	0.77	96	1.54	96	3.21	104
5	<LOQ	0.76	95	1.51	94	3.19	103
Average		0.77	96	1.51	94	3.2	102.8
CV%		5.1		2.3		2.3	

Samples spiked with 0.80, 1.60, or 3.10 of ZVNS1 were diluted 2 times and measured by the 32/D6 ELISA. CV%: percent coefficient of variation.

**Table 3 biomolecules-10-01652-t003:** Interday precision of the test with ZVNS1-spiked serum.

Day	Unspiked Serum	Spiking 0.80 ng/mL		Spiking 1.60 ng/mL		Spiking 3.10 ng/mL	
		Measured (ng/mL)	Recovery %	Measured (ng/mL)	Recovery %	Measured (ng/mL)	Recovery %
1	<LOQ	0.77	97	1.56	104	3.02	97
2	<LOQ	0.67	84	1.38	86	2.84	92
3	<LOQ	0.65	82	1.64	106	3.22	104
4	<LOQ	0.58	73	1.48	92	2.97	96
5	<LOQ	0.77	98	1.51	94	3.19	103
Average		0.69	86.8	1.51	96.4	3.05	98.4
CV%		8.1		9.8		16.0	

Samples spiked with 0.80, 1.60, or 3.10 of ZVNS1 were diluted 2 times and measured by the 32/D6 ELISA.

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
