# Peer review of "Highly Sensitive Detection of Zika Virus Nonstructural Protein 1 in Serum Samples by a Two-Site Nanobody ELISA"

_biomolecules, 2020, doi:10.3390/biom10121652_

Round 1

Reviewer 1 Report

In this work, the authors describe the identification of multiple nanobodies against the zika virus NS1 protein. A good pair of Nbs ws selected to develop a sensitive and specific double nanobody sandwich ELISA to screen human sera for current Zika infections.

The work is nicely developed and interesting. Nevertheless, the work is not that original or novel in the sense that all techniques were already in place, although the Nanobody pair selection technique for sandwich ELSIA was not yet used for the Zika viral NS1 protein. 

There are a few minor text editing issues that need the authors attention:

Lines 33-40 should be removed as those are guidelines to authors.

Line 54: since  RT PCR can be real time PCR and Reverse transcription PCR, it might be better to use Quantitative PCR 

Line 114: make sure it is 10E7 (7 in superscript)

Same remark on line 144: 10E11

and line 209: 10E8 and 10E7

Line 148 'E.coli' should be in italic

It might be preferred to show the EILSA signals in 3 bars next to each other instead of cumulative. Although this is obviously a personal preference and I leave out to the authors to judge. (If the figure becomes too crowded, they could show a few selected clones in Figure 1 (and bring the remaining clones in a supplementary figure. 

Lines 164 and 166 H202 and H2SO4 should have numbers in subscripts

More important: could the authors explain teh numbering of clones in going from Fig 1 to figure 2 and remaining parts. Foe example clone 210 and 212 are which clones in figure 1?

Line 330: Don't think it is justified to claim that there is a linear range from 0 to 6.25 ng/mL. For sure it is not 0. 

Furthermore, Figure 9 left and right are these data from teh same experiment? in the left panel the linear range seems to go from 2 to 10 ng/mL, while in the right panel it is more like 1 to 6 ng/mL (if you add data for next datapoint (10 is this off the linear ?)

Line340 noticed ('d' should be added) 

Author Response

Reviewer 1

In this work, the authors describe the identification of multiple nanobodies against the zika virus NS1 protein. A good pair of Nbs ws selected to develop a sensitive and specific double nanobody sandwich ELISA to screen human sera for current Zika infections.

The work is nicely developed and interesting. Nevertheless, the work is not that original or novel in the sense that all techniques were already in place, although the Nanobody pair selection technique for sandwich ELSIA was not yet used for the Zika viral NS1 protein.

There are a few minor text editing issues that need the authors attention:

Lines 33-40 should be removed as those are guidelines to authors.

Response: Done

Line 54: since RT PCR can be real time PCR and Reverse transcription PCR, it might be better to use Quantitative PCR

Response: Done

Line 114: make sure it is 10E7 (7 in superscript)

Response: Done

Same remark on line 144: 10E11

Response: Done

and line 209: 10E8 and 10E7

Response: Done

Line 148 'E.coli' should be in italic

Response: Done

It might be preferred to show the EILSA signals in 3 bars next to each other instead of cumulative. Although this is obviously a personal preference and I leave out to the authors to judge. (If the figure becomes too crowded, they could show a few selected clones in Figure 1 (and bring the remaining clones in a supplementary figure.

Response: We understand the preference of the reviewer, but as he/she kindly suggests, we prefer to leave it as is presented in the manuscript.

Lines 164 and 166 H202 and H2SO4 should have numbers in subscripts

Response: Done

More important: could the authors explain teh numbering of clones in going from Fig 1 to figure 2 and remaining parts. Foe example clone 210 and 212 are which clones in figure 1?

Response: The correlation of the numbers used in figure 1 and the name of clones was added to figure S3 y S4.

Line 330: Don't think it is justified to claim that there is a linear range from 0 to 6.25 ng/mL. For sure it is not 0.

Furthermore, Figure 9 left and right are these data from teh same experiment? in the left panel the linear range seems to go from 2 to 10 ng/mL, while in the right panel it is more like 1 to 6 ng/mL (if you add data for next datapoint (10 is this off the linear ?)

Response: The reviewer is right, and the linear range was modified in the text to 0.2-6.25 ng/mL. The inclusion of the point corresponding 12.5 ng/mL brings the R2  from 0.991 to 0.977 and thus we did not include it.

Reviewer 2 Report

In their study Delfin-Riela et al describe the selection of nanobodies against the non-structural protein 1 (NS1) of Zika virus and their application to capture and detect NS1 in a sandwich immunoassay with the aim to generate a robust and sensitive assay to detect NS1 as secreted biomarker for viral infection in serum samples of infected individuals. While their approach to generate ZIKA NS1 specific nanobodies is straight forward and successful this study still comprises severe shortcomings which needs to be addressed before it can be considered for publication in Biomolecules.

Major comments:

In their study the authors show the performance of preselected nanobody-pairs to detect spiked in NS1 in serum samples from healthy individuals. To demonstrate the significance of their findings detailed benchmark studies are needed. Most importantly relevant patient derived serum samples should be included and the authors should compare their nanobody-based sandwich immunoassay with either well established diagnostic assays using RT-PCR (e.g. as described by Pyke et al 2014 http://dx.doi.org/10.1371/currents.outbreaks.4635a54dbffba2156fb2fd76dc49f65e) or with more recently developed antibody-based assays (e.g. as described by Bosch et al.,2017, DOI: 10.1126/scitranslmed.aan1). Important information on the selected nanobodies such as recognized epitopes / domains, affinities, stabilities are missing.

Minor comments:

Line 33 - 40: The general guidelines how to write an introduction should be removed

Line 109: The authors should provide a permission no. of the ethical committee which is clearly linked to the permission in conducting animal experiments

Figure 1: The author should describe in which format the nanobodies were tested here (soluble extract, purified etc.). No standard deviations are shown. What is the no. of biological/technical replicates?

Figure 2 and 3 : Same comment as for Figure 1. To me both figures are highly redundant, thus one of them should be moved to Suppl.Info

Figure 5 / Figure 7 :Cross reactivity to NS1 of other flavivirus should be tested in solution not on absorbed proteins. A more detailed experiment using the finally selected Nb-pairs in a capture/detection array should be shown

Author Response

Reviewer 2

In their study Delfin-Riela et al describe the selection of nanobodies against the non-structural protein 1 (NS1) of Zika virus and their application to capture and detect NS1 in a sandwich immunoassay with the aim to generate a robust and sensitive assay to detect NS1 as secreted biomarker for viral infection in serum samples of infected individuals. While their approach to generate ZIKA NS1 specific nanobodies is straight forward and successful this study still comprises severe shortcomings which needs to be addressed before it can be considered for publication in Biomolecules.

Major comments:

In their study the authors show the performance of preselected nanobody-pairs to detect spiked in NS1 in serum samples from healthy individuals. To demonstrate the significance of their findings detailed benchmark studies are needed. Most importantly relevant patient derived serum samples should be included and the authors should compare their nanobody-based sandwich immunoassay with either well established diagnostic assays using RT-PCR (e.g. as described by Pyke et al 2014 http://dx.doi.org/10.1371/currents.outbreaks.4635a54dbffba2156fb2fd76dc49f65e) or with more recently developed antibody-based assays (e.g. as described by Bosch et al., 2017, DOI: 10.1126/scitranslmed.aan1). Important information on the selected nanobodies such as recognized epitopes / domains, affinities, stabilities are missing.

Response: In our experience, the diagnostic performance of any test, ultimately relies on its analytical robustness which should be demonstrated first. Therefore, the main aim of our study was the development of a sensitive and robust method for the detection of ZIKV NS1 in serum samples that could be reproduced in any laboratory and be the basis for future diagnostic tests.  In that sense, the focus of our report was the careful selection of the nanobody pair used for detection of NS1, and the analytical validation of the test in the target matrix, in terms of sensitivity, specificity, accuracy and reproducibility. Anyway, we definitively agree with the reviewer that the diagnostic use of the method will require its validation with a large serum collection, and that study is undergoing through an international collaboration (though seriously delayed due to the current travel and serum exchange limitations imposed by the COVID-19 pandemic). To clarify this point, we have now modified the Conclusions section as follows: “Considering that these Nbs can be fully reproduced at very low cost from their amino acid sequences (provided in this study), the ZVNS1-32/D6 ELISA could be replicated in any laboratory, and be the basis for an affordable tool for early diagnosis of ZIKV. Nevertheless, a validation process with relevant patient serum samples should be conducted locally in order to demonstrate its diagnostic value.”

Regarding the comparison with other methods, as suggested by the reviewer, we have now added information about this point in Conclusions as follows: “The limit of quantification of the test in serum, 0.80 ng/mL, was established experimentally studying the recovery (accuracy) and reproducibility with a panel of spiked serum samples. This sensitivity compares favorably with other well established antibody-based assays for NS1, such as the one described by Bosch et al. that was able to detect up to 18 ng/mL [28].

Minor comments:

Line 33 - 40: The general guidelines how to write an introduction should be removed

Response: Done

Line 109: The authors should provide a permission no. of the ethical committee which is clearly linked to the permission in conducting animal experiments

Response: The permission number corresponding to the protocol for conducting the animal immunization (CEUA-1-141107) was added in section 2.1. (line 101).

Figure 1: The author should describe in which format the nanobodies were tested here (soluble extract, purified etc.). No standard deviations are shown. What is the no. of biological/technical replicates?

Response: The nanobodies were tested as supernatants. This information was now added in the caption of figure 1. This was a large initial qualitative screening of hundreds of supernatants and was performed by single measurements. This is now explained in the text.

Figure 2 and 3: Same comment as for Figure 1. To me both figures are highly redundant, thus one of them should be moved to Suppl.Info

Response: The nanobodies were tested as supernatants, which were performed in duplicates. This information was now added in the caption of figure. Figure 2 was moved to Supplementary Information as figure S2.

Figure 5 / Figure 7: Cross reactivity to NS1 of other flavivirus should be tested in solution not on absorbed proteins. A more detailed experiment using the finally selected Nb-pairs in a capture/detection array should be shown

Response: The experiment suggested by the reviewer was done. As expected no cross-reactivity was observed with any of the other flavivirus proteins. The result was included in the first paragraph of section 3.4

Round 2

Reviewer 2 Report

Major

As previously mentioned, I would like to see more relevant data from real patient samples to demonstrate the functionality of the assay. Alternatively the authors might consider to include a direct comparision using an antibody-based ELISA (e.g. as used in Mesci et al., SciRep 2018, DOI:10.1038/s41598-018-19526-4)

This would strengenth the importance of the described findings which might be important, as the description of the Nb selection is rather trivial and no detailed characterization of the identified Nbs regarding stability, affinity, epitope etc is provided

Minor

The authors should carefully check the wording. Especially they should not mix  up the term "nanobody" with "antibody" (linea 279, 282, 291)

Author Response

Reviewer 2

As previously mentioned, I would like to see more relevant data from real patient samples to demonstrate the functionality of the assay. Alternatively the authors might consider to include a direct comparision using an antibody-based ELISA (e.g. as used in Mesci et al., SciRep 2018, DOI:10.1038/s41598-018-19526-4)

This would strengenth the importance of the described findings which might be important, as the description of the Nb selection is rather trivial and no detailed characterization of the identified Nbs regarding stability, affinity, epitope etc is provided

Response: The Editor said “While the addition of clinical samples would increase the impact of the paper, we follow the reasoning of reviewer 1 that the addition of this experiment is not necessary for publication in this special issue.”

Minor

The authors should carefully check the wording. Especially they should not mix  up the term "nanobody" with "antibody" (linea 279, 282, 291)

Response: This has been corrected as suggested by the Reviewer.